Identification and validation of candidate genes dysregulated in alveolar macrophages of acute respiratory distress syndrome

Mao Yong 1
Lv Xin lvxinshanghai@163.com 1
Xu Wei 1
Ying Youguo 2
Qin Zonghe 1
Liao Handi 1
Chen Li 1
Liu Ya 1
1 Shanghai Ninth People’s Hospital, Shanghai Jiao Tong University School of Medicine, Department of Intensive Care Unit , Shanghai , China
2 Shanghai Ninth People’s Hospital, Shanghai Jiao Tong University School of Medicine, Emergency Department , Shanghai , China
Young Howard
Electronic publication date: 2021 Oct 26
Publication date: 2021
Volume: 9
Electronic Location ID: e12312
Received 2021 May 31; Accepted 2021 Sep 24
Copyright: ©2021 Mao et al.
Copyright year: 2021
Copyright holder: Mao et al.
License: This is an open access article distributed under the terms of the Creative Commons Attribution License, which permits unrestricted use, distribution, reproduction and adaptation in any medium and for any purpose provided that it is properly attributed. For attribution, the original author(s), title, publication source (PeerJ) and either DOI or URL of the article must be cited.
License URL: https://creativecommons.org/licenses/by/4.0/

Keywords: ARDS, Macrophage, FPR3, CCR2, Chemotaxis

Funding: Ninth People’s Hospital JYZZ036 This work was supported by Fundamental research program funding of Ninth People’s Hospital affiliated to Shanghai Jiao Tong university of School of Medicine (No. JYZZ036). The funders had no role in study design, data collection and analysis, decision to publish, or preparation of the manuscript.

==============================
Background

Acute respiratory distress syndrome (ARDS) is a common cause of death in ICU patients and its underlying mechanism remains unclear, which leads to its high mortality rate. This study aimed to identify candidate genes potentially implicating in the pathogenesis of ARDS and provide novel therapeutic targets.

Methods

Using bioinformatics tools, we searched for differentially expressed genes (DEGs) in an ARDS microarray dataset downloaded from the Gene Expression Omnibus (GEO) database. Afterwards, functional enrichment analysis of GO, KEGG, GSEA and WGCNA were carried out to investigate the potential involvement of these DEGs. Moreover, the Protein–protein interaction (PPI) network was constructed and molecular complexes and hub genes were identified, followed by prognosis analysis of the hub genes. Further, we performed qRT-PCR, Western Blot and flow cytometry analysis to detect candidate genes of CCR2 and FPR3 in macrophage model of LPS-induced ARDS and primary alveolar macrophages(AMs). Macrophage chemotaxis was evaluated using Transwell assay.

Results

DEGs mainly involved in myeloid leukocyte activation, cell chemotaxis, adenylate cyclase-modulating G protein-coupled receptor signaling pathway and cytokine-cytokine receptor interaction. Basing on the constructed PPI network, we identified five molecular complexes and 10 hub genes potentially participating in the pathogenesis of ARDS. It was observed that candidate genes of CCR2 and FPR3 were significantly over-expressed in primary alveolar macrophages from ARDS patients and macrophgae model of LPS-induced ARDS. Moreover, in vitro transwell assay demonstrated that CCR2 and FPR3 down-regulation, respectively, inhibited LPS-triggered macrophage chemotaxis toward CCL2. Finally, a positive correlation between FPR3 and CCR2 expression was confirmed using pearson correlation analysis and Western Blot assay.

Conclusions

Our study identified CCR2 and FPR3 as the candidate genes which can promote macrophage chemotaxis through a possible interaction between FPR3 and CCL2/CCR2 axis and provided novel insights into ARDS pathogenesis.

Introduction

ARDS is a prevailing death cause of ICU patients. As a form of non-cardiogenic pulmonary oedema, ARDS is characterized by diffusive alveolar damage, followed hyaline-membrane formation, epithelial-cell hyperplasia and interstitial edema. These pathological changes are secondary to inflammation process mainly caused by severe infection, trauma, burning and massive transfusion.

Although there have been several therapeutic strategies such as lung-protective ventilation (LPV), prone positioning and inhaled nitric oxide, the current therapy for ARDS is inadequate. A recent study indicates a high mortality rate of 46% for severe ARDS patients (Bellani et al., 2016) and patients who survive ARDS are likely to suffer from cognitive decline, depression, persistent skeletal-muscle weakness and so on (Herridge et al., 2016; Herridge et al., 2011). However, effective therapeutic targets have been unavailable so far and the molecular mechanism underlying the pathogenesis of ALI/ARDS remains unknown.

As lung’s initial response to injury, resident alveolar macrophages secret pro-inflammatory cytokines and chemokines, inducing neutrophils and monocytes to migrate and accumulate into alveolar spaces. Activated neutrophils further release toxic mediators, induce tissue damage and lead barrier disruption. Meanwhile, alveolar epithelial cells and effector T cells are activated to promote and sustain inflammation and tissue damage. Besides, aberrant endothelial activation also contributes to the barrier disruption (Aggarwal, King & D’Alessio, 2014). Although a variety of target and effector cells are believed to be associated with the initiation and progress of ALI/ARDS, accumulating studies have shown that macrophages are the key orchestrators.

To explore the molecular mechanism underlying the pathogenesis of ALI/ARDS, we used data mining techniques to investigate the potential genes responsible for the activation and recruitment of alveolar macrophages in ARDS. Here, we used microarray dataset GSE89953 created by Morrell et al. (2018) to conduct a genome-wide gene expression analysis aiming to investigate the DEGs between alveolar macrophages and peripheral blood monocytes (PBMs) in ARDS patients. Afterwards, we performed functional enrichment analysis of DEGs to reveal the biologic programs and pathways which may participate in the activation and migration of macrophages in ARDS. Furthermore, the protein-protein interaction (PPI) network was constructed and the top 10 hub genes of GNG2, S1PR3, CCR2, CX3CR1, GPER1, CXCL16, CCL13, FPR3, P2RY13 and MAPK1 were selected for prognostic analysis using mRNA expression data of GSE116560 (Morrell et al., 2019).

Next, we aimed to validate the expression trend of the hub genes obtained from microarray assay in an in vitro experiment. To this end, THP-1 cells were differentiated into macrophages using phorbol 12-myristate 13-acetate (PMA) and macrophage model of LPS-induced ARDS was established. Then, we performed qRT-PCR and Western blot and revealed that two hub genes, FPR3 and CCR2, whose mRNA and protein levels were significantly elevated after LPS stimulation in a time-dependent manner, may play a crucial role in the activation and accumulation of macrophages in ARDS. Besides, we isolated primary alveolar macrophages from bronchoalveolar lavage fluid (BALF) collected from ARDS patients using magnetic-activated cell sorting (MACS). Likewise, mRNA and protein levels of CCR2 and FPR3 were substantially elevated in primary alveolar macrophages of ARDS patients. Afterwards, flow cytometry analysis indicated a markedly enhanced CCR2 expression on surface of primary alveolar macrophages from ARDS subjects and macrophages of LPS-induced ARDS. However, FPR3 was hardly detected mainly due to its localization to intracellular vesicles after translation.

Both FPR3 and CCR2 are members of seven transmembrane domains, G protein-coupled receptors (GPCRs) which include two major subfamilies: “classical GPCRs” and “chemokine GPCRs”. GPCRs can recognize pathogen- or damage-associated chemotactic molecular patterns (PAMPs and DAMPs) and mediate leukocyte infiltration to disease sites, which is considered as a hallmark of inflammation. Formyl-peptide receptors (FPRs) belong to the “classical GPCRs family”. In human, there are 3 FPRs, FPR1, FPR2 and FPR3. FPR1 and FPR2 express in a wide range of tissues, organs and certain malignant tumor cells and involve multiple biological functions such as leukocyte trafficking, wound healing and carcinogenesis. Unlike FPR1 and FPR2, FPR3 merely expresses in monocytes and dendritic cells (DCs) and is highly phosphorylated and more localized to intracellular vesicles. Currently, the overall function of FPR3 remains unclear. Known as a major chemokine receptor on macrophages and monocytes , CCR2 has several high-affinity ligands including MCP-1(CCL2), MCP-2(CCL7), MCP-3 (CCL8), MCP-4 (CCL13) and MCP-5 (CCL12). Growing evidence suggests that aberrant CCR2 activation by monocyte chemoattractant proteins can induce atherosclerosis (Colin, Chinetti-Gbaguidi & Staels, 2014), insulin resistance (Bai & Sun, 2015) and several types of malignant tumor (Qian et al., 2011; Qian & Pollard, 2010; Ueno et al., 2000). However, the actual role of CCR2 in the initiation and progression of ARDS is still unknown.

CCL2 has been verified as a key chemotactic factor released in bronchoalveolar lavage of a murine model of LPS-induced ALI (Li et al., 2016). Meanwhile, a substantial overexpression of CCR2 was confirmed in macrophage model of LPS-induced ARDS and primary alveolar macrophages. These findings have led to an interest in a potential role of CCL2/CCR2 axis in ARDS pathogenesis. Accumulated evidence supports the notion that CCL2 and its receptor CCR2 participate in cancer metastasis, inflammation disease, obesity and atherosclerosis. However, its actual role in ARDS is still undefined.

Using an in vitro migration assay, we demonstrated that LPS stimulation significantly enhanced chemotaxis of PMA-differentiated macrophages toward chemokine CCL2 and this effect can be abrogated by CCR2 silence, which suggested that CCL2/CCR2 axis may function in regulating macrophage chemotaxis in ARDS. Moreover, we revealed that LPS-triggered macrophage chemotaxis toward CCL2, but not toward fMLP, was substantially attenuated by siRNA-FPR3 transfection. This suggested a possible regulatory effect of FPR3 on CCL2/CCR2 axis. Afterwards, we performed pearson correlation analysis with mRNA expression data of GSE89953 and GSE116560 and revealed a strong correlation of FPR3 with CCR2 expression in alveolar macrophages of ARDS patients. Furthermore, siRNA-mediated FPR3 silence remarkably inhibited LPS-triggered CCR2 elevation on macrophages, while siRNA-CCR2 transfection had no effects on FPR3 expression. These findings suggested that FPR3 may function as a potential chemotactic mediator by modulating CCL2/CCR2 axis. Our results may help to delineate a novel insight into the pathogenesis of ARDS and lay a foundation for subsequent functional studies of CCR2 and FPR3 protein.

Materials & Methods

Ethics statement

This study was approved by the ethics committee of Shanghai Ninth People’s Hospital, Shanghai Jiao Tong University School of Medicine. Written informed consent was obtained from all patients and volunteers.

Data collection and study design

Two mRNA expression datasets of ARDS (GSE89953 and GSE116560) (Morrell et al., 2019; Morrell et al., 2018) were downloaded from the gene expression omnibus (GEO, https://www.ncbi.nlm.nih.gov/geo) database. GSE89953 included 26 alveolar macrophage samples isolated from bronchoalveolar lavage fluid of ARDS patients and paired 26 peripheral blood monocyte samples. As for GSE116560, 68 alveolar macrophage samples were collected from ARDS patients and 30 of 68 were obtained on ARDS onset day 1. The above datasets were detected using GPL6883 platforms and natural scale quantile normalization was performed.

As the initial response to injury, the exudative phase of ARDS is characterized by alveolar macrophage activation inducing the release of pro-inflammatory mediators and chemokines that promote the accumulation of monocytes and neutrophils. Therefore, alveolar macrophages which were derived and differentiated from peripheral blood monocytes were considered as the main contributor to alveolar epithelial and endothelial damage. To clarify the molecular mechanism underlying alveolar macrophage activation and accumulation in the early phase of ARDS, mRNA expression data of GSE89953 was used to screen and identify the DEGs in alveolar macrophages of ARDS. Moreover, we utilized GSE116560 to conduct prognostic analysis of the hub genes obtained from the GSE89953 dataset.

DEGs screening and identification

To investigate the underlying molecular mechanism of macrophage activation and infiltration into pulmonary alveoli in the early stage of ARDS, mRNA expression dataset (GES89953) was downloaded and differential mRNA expression in alveolar macrophage relative to peripheral blood monocyte was acquired using the limma package in R language. P < 0.01 and —log2fold change (FC)—≥1 was set as threshold values to define DEGs.

Principal component analysis

Principal component analysis (PCA) is a multivariate statistical method that identifies patterns and classifies the factors which influence a certain phenomenon. As a technique widely used in medical field, PCA decomposes a set of correlated variables into a set of uncorrelated variables named as principal components (PCs) and reduces dimensionality by discarding the less important principal components. In this study, we used the Principal Component Analysis tool of Origin software to perform PCA basing on the mRNA expression matrix of the top 300 DEGs across the 26 alveolar macrophage and paired peripheral blood monocyte samples of ARDS patients. Since 52 samples were included, 52 PCs were generated.

Pathway and process enrichment analysis

Metascape (http://metascape.org) is an efficient, free, web-based and user-friendly analysis tool for experimental biologists to comprehensively analyze and interpret OMICs-based studies (Zhou et al., 2019). In this study, Metascape was chosen to conduct process and pathway enrichment analysis for the top 300 DEGs. For this purpose, the Gene Ontology (GO) terms for biological process and Kyoto Encyclopedia of Genes and Genomes (KEGG) pathways were enriched using the Metascape analysis tool. P-values <0.01, a minimum count of 3 and enrichment factor >1.5 were chosen as cutoff values and remaining significant terms were grouped into clusters based on Kappa-statistical similarities. Specifically, Kappa score of 0.3 was applied as the threshold and terms with a similarity >0.3 were considered as a cluster. The term which was most statistically significant within a cluster was selected to represent this cluster.

Next, Gene Set Enrichment Analysis (GSEA) (Subramanian et al., 2005) was performed basing on expression data of GSE89953 to explore the potential biological processes and pathways involved in the activation and accumulation of macrophages in ARDS. GO and KEGG gene sets were chosen as priori gene sets for functional enrichment analysis.

Weighted gene co-expression network analysis and module identification

Weighted gene co-expression network analysis (WGCNA) was performed using iDEP.91 web application (https://bioinformatics.sdstate.edu/idep/) (Ge, Son & Yao, 2018) and applied to construct gene co-expression networks based on mRNA expression matrix (GSE89953). The processes of WGCNA mainly included several steps. Firstly, a gene-gene similarity network was established and, afterwards, divided into network modules basing on expression similarity of group genes. Thirdly, the correlation between modules and phenotypic traits was established and, finally, the “driver” genes in modules were identified. β value of 14 was chosen as soft threshold on the basis of scale independence and mean connectivity plots.

Protein–protein interaction analysis

A protein–protein interaction network of the top 300 DEGs was constructed through the STRING database (http://string-db.org/) (Szklarczyk et al., 2019) with a confidence score >0.7 and the PPI network was visualized by Cytoscape (Version 3.7.2) software. The module analysis of the PPI network was performed using the Molecular Complex Detection (MCODE) of Cytoscape and functional enrichment analysis of the obtained modules were carried out using Metascape tool. Moreover, hub genes were screened out using cytoHubba tool of Cytoscape calculating by MCC method and the top 10 hub genes were chosen for further analysis.

Prognostic analysis of the hub genes

mRNA expression matrix of GSE116560 was downloaded from GEO database and the 30 alveolar macrophage samples gathered on ARDS onset day 1 were selected for prognostic analysis. Ventilation-free days (VFDs) was a common description for clinical outcomes of ARDS patients. In this study, VFDs > 0 was defined as subjects who were alive and liberated from mechanical ventilation within 28 days and VFDs = 0 was subjects who died or were persistently supported on mechanical ventilation. For these 30 ARDS patients whose alveolar macrophage samples were available on the first day, the VFDs ranged from 0 to 25 and the average value of 12 was set as the cutoff value. The patients whose VFDs exceeded 12 were defined as individuals with better clinical outcomes and those with VFDs < 12 were accepted as poor prognosis individuals. T test was used to investigate the association between transcript levels of the hub genes and the clinical outcomes and P < 0.05 was defined as statistically significant.

Cell culture and treatment

The human monocytic cell line THP-1 was purchased from the American Type Culture Collection (Manassas, VA, USA) and cultured in RPMI-1640 medium with 10% fetal bovine serum, 100 U/ml penicillin and 100 µg/ml streptomycin at 37 °C in an incubator with a humidified atmosphere containing 5% CO2.

1 × 106 THP-1 cells were seeded into 6-well plate and differentiated with PMA (Sigma-Aldrich, St Louis, MO, USA) for 48 h in a concentration of 50ng/ml. Cell adhesion, spreading and decreased nucleocytoplasmic ratio, as the main features for macrophage differentiation, were visualized by phase-contrast microscopy. Surface markers of CD14 (monocyte marker) and CD68 (macrophage marker) during differentiation were detected by flow cytometry. To measure the response of PMA-differentiated macrophages to LPS (Escherichia coli 055:B5; Sigma-Aldrich, St Louis, MO, USA), cells were exposed to LPS (1 µg /ml) for 12 h after a recovery period (an incubation of 12 h in fresh RPM-1640 removing PMA and 10% FBS) and levels of TNFα in cell supernatants were quantified using Human TNFα ELISA Kit (Sangon Biotech, Shanghai, China) according to the manufacturer’s instructions. After the acquisition of a macrophage-like phenotype, cells were treated with LPS (1 µg /ml) for 0, 1, 3, 6, 12 and 18 h, respectively, and macrophage model of LPS-induced ARDS was successfully established.

Primary alveolar macrophages isolation

BALF samples were collected from six mechanically ventilated patients diagnosed as ARDS according to the Berlin Definition (ARDS Definition Task Force et al., 2012). Samples were collected within 3 days after-diagnosis and, meanwhile, four BALF samples were obtained from healthy volunteers as negative controls. Magnetic-activated cell sorting was performed and negative selection for alveolar macrophages was achieved by incubating cells collected from the BALF samples with antibody-conjugated microbeads specific for the following markers: CD3 (T cells), CD15 (neutrophils), CD19 (B cells), CD235a (red blood cells), CD229f (eosinophils, basophils) and CD326 (epithelial cells).

RNA isolation and qRT-PCR

Total RNA extraction from cell samples was performed using TRIzol reagent (Thermofisher Scientific, Waltham, MA, USA) according to the manufacturer’s instructions. Primers were synthesized by Sangon Biotech and the sequences were shown in Table S1. The reverse-transcription was performed using PrimeScript™ RT Master Mix (Takara, Tokyo, Japan). The mRNA expression levels of 6 hub genes (CCL13, FPR3, PLAU, CX3CR1, CXCL16 and CCR2) in cells were detected by qPCR using ChamQ Universal SYBR qPCR Master Mix (Vazyme, Nanjing, China) and calculated as described (Yu et al., 2016). GAPDH mRNA was used as the internal reference and the relative mRNA expressions were calculated using the 2−ΔΔCT method.

Western blot

Cells were lysed using RIPA lysis buffer (Beyotime, Shanghai, China) in the presence of Cocktail protease inhibitor (Abmole, USA) and Cocktail phophatase inhibitor (Abmole, USA). Lysate protein concentrations were determined by BCA Protein Assay Kit (Beyotime, Shanghai, China). Equivalent protein from different samples was separated by 10% SDS-PAGE protein electrophoresis and the separated protein samples were transferred onto the Immobilon PVDF membranes. Then, the membranes were incubated with 5% skim milk to block the non-specific sites. Rabbit polyclonal anti-human FPR3 antibody (orb608040, Biorbyt, UK), rabbit polyclonal anti-human CCR2 antibody (NBP1-48337, Novus Biologicals, USA) and HRP-conjugated secondary antibody (1:2000, Proteintech, IL, USA) were used. GAPDH (1:5000, Proteintech, IL, USA) was simultaneously used as an internal control. Signals were detected and visualized with Immobilon Western Chemiluminescent HRP Substrate (Merck Millipore, Darmstadt, Germany).

Flow cytometry analysis

THP-1-derived macrophages or primary alveolar macrophages collected from ARDS patients were resuspended in 50 µl of PBS. Cell suspension was incubated with 2 µl of APC anti-human CD14 antibody (Biolegend, USA) , FITC anti-human CD68 antibody (Biolegend, USA) and PE anti-human CCR2 antibody (Biolegend, USA) in dark at 4 °C for 30 mins. Flow cytometry analysis was carried out with BD LSRFortessa™ X-20 cell analyzer (BD Biosciences, USA).

Chemotaxis assay

THP-1 cells were differentiated into macrophages using PMA as described above. siRNA-FPR3, siRNA-CCR2 and siRNA-control were purchased from Fubio Biological Technology (Shanghai, China) and transiently transfected into PMA-differentiated macrophages. At 18 h post-transfection with siRNA-FPR3, siRNA-CCR2 or siRNA-control, macrophages were added into the upper chamber of Transwell plate (24-well plates, 8.0 µm pores, Corning, USA) in 100 µl serum-free medium containing LPS (1 µg /ml) and allowed to migrate toward CCL2 (100 ng/ml) (300-39, Peprotech, USA), fMLP (10−8M) (F3506, Sigma-Aldrich, St Louis, MO, USA) or culture medium in the lower chamber. After a 3 h incubation at 5%CO2 and 37° C, non-migrated macrophages (upper chamber) and migrated macrophages (lower chamber) were collected and counted.

Statistical analysis

All data were presented as mean ± SD. Statistical analysis was performed with GraphPad software version 6.01. Comparison between 2 groups was carried out using unpaired, 2-tailed, Student’s t test. When there were more than 2 groups, one-way AVOVA was performed, followed by Bonferroni correction when necessary. Differences were considered to be statistically significant when the p-value was less than 0.05.

Results

Screening and identification of differentially expressed genes in alveolar macrophages of ARDS patients

To identify the DEGs in alveolar macrophages of ARDS patients, mRNA expression data (GSE89953) was downloaded from GEO database. Variance stabilization and quantile normalization was performed for the raw microarray data and screening and identification of the DEGs in ARDS alveolar macrophages compared to peripheral blood monocytes was accomplished using the limma package of R language. According to the thresholds set (P < 0.01 and —log2FC— ≥ 1), 385 up-regulated and 325 down-regulated genes were found respectively (Fig. 1A). Subsequently, we selected the top 300 DEGs ranked in ascending order of adjusted p-value for further analysis. Basing on these 300 DEGs, heatmap was generated using the online TBtools software (Chen et al., 2020) and revealed a remarkable disparity of transcriptional pattern between alveolar macrophages and peripheral blood monocytes in ARDS patients (Fig. 1B).

Figure 1 Identification of DEGs between alveolar macrophages and peripheral blood monocytes.

(A) Volcano plot of DEGs in GSE89953. Purple and green dots represent up-regulated and down-regulated genes, respectively. Volcano plot shows all DEGs. P < 0.01 and —log2fold change (FC)— ≥ 1 is set as threshold values for DEGs. (B) Heatmap plotting is based on the top 300 DEGs. Each column represents a sample and each row represents a DEG. The ascending normalized expression level in the heatmap is represented by the color and size of dots. A larger size and darker red dot means gene up-regulation. Conversely, a smaller size and lighter red one indicates gene down-regulation. (C) Principal component analysis basing on the expression matrix of the top 300 DEGs across the 26 alveolar macrophage and paired peripheral blood monocyte samples from ARDS patients.

Principle component analysis (PCA)

The PCA results indicated that 3 of the 52 PCs correspond to 81.2% of the overall variance. More specifically, principal component 1 and principal component 2 accounted for 74.6% and 4.1% of overall variance, respectively. As shown in Fig. 1C, a clear segregation of gene expression was revealed between alveolar macrophages and peripheral blood monocytes in ARDS patients.

Pathway and process enrichment analysis

Process and pathway enrichment analysis were performed using Metascape analysis tool to predict the potential biological processes and pathways participating in ARDS onset. As shown in Fig. 2A, the most enriched GO terms for biological process were ‘myeloid leukocyte activation’, ‘cell chemotaxis’, ‘regulation of inflammatory response’, ‘positive regulation of hydrolase activity’ ,‘lymphocyte activation’ , ‘positive regulation of defense response’, ‘leukocyte differentiation’, ‘response to wounding’, ‘organic hydroxy compound metabolic process’, ‘phagocytosis’ and ‘regulation of cytokine production’. Meanwhile, KEGG analysis indicated that the top 300 DEGs significantly enriched in pathways of ‘Phagosome’, ‘Apoptosis’, ‘B cell receptor signaling pathway’, ‘Chemokine signaling pathway’ and ‘PPAR signaling pathway’.

Figure 2 Functional enrichment analysis of DEGs.

(A) GO and KEGG pathway analysis of the top 300 DEGs using Metascape tool. The y axis represents GO and KEGG categories and the x axis represents the enrichment score. The orange columns indicate GO analysis and the red ones mean KEGG pathway analysis. (B) Clustering dendrogram of DEGs. Eight gene modules are colored by turquoise, blue, brown, yellow, green, red, black and pink, respectively. (C) GO analysis of the eight modules. Likewise, the y axis represents GO terms and the x axis represents the enrichment score. Each module is marked by a color as described above. (D) KEGG pathway analysis of the eight modules. The y axis indicates KEGG categories and the x axis means the enrichment score. As mentioned above, each color symbolizes a certain module.

GSEA indicated that up-regulated genes in alveolar macrophages statistically enriched in GO and KEGG gene sets of ‘Regulation of Non Canonical Wnt Signaling Pathway’, ‘Monocyte Chemotaxis’, ‘Adenylate Cyclase inhibiting G protein Coupled Receptor Signaling Pathway’, ‘Macrophage Migration’, ‘ERK1 and ERK2 Cascade’, ‘Negative Regulation of Apoptotic Signaling Pathway’, ‘Cytokine-Cytokine Receptor Interaction’, ‘Regulation of Autophagy’, ‘PPAR Signaling Pathway’, ‘Complement and Coagulation Cascades’ and ‘JAK-STAT Signaling Pathway’, which was consistent with the results obtained from Metascape tool.

Gene co-expression networks identification and enrichment analysis of modules

Lowly expressed genes were filtered out and 819 remaining genes were used in WGCNA analysis. According to the criterion of scale free topology fix index reaching 0.9 and an appropriate mean connectivity value, the soft threshold value was set as 14. Finally, eight co-expression gene modules were identified by the dynamic tree cut method and each module was marked by a color. The turquoise was the largest module with 422 genes and the pink was the smallest with 23 genes. The clustering dendrogram of genes was displayed in Fig. 2B.

Biological process and pathway enrichment analysis of the eight modules were performed using Metascape tool as previously described and a substantial proportion of the resulting terms shared in common with those obtained from previous analysis. As shown in Fig. 2C, genes of module 1, 2, 6 and 8 mainly involved in immune response, leukocyte activation and response to external stimulus. Besides, module 3 probably participated in cholesterol mechanism. As for module 4, genes significantly enriched in neutrophil chemotaxis and migration. Meanwhile, KEGG analysis indicated a significant enrichment in pathways of ‘complement and coagulation cascades’, ‘cytokine-cytokine receptor interaction’, ‘Cell adhesion molecules’, ‘MAPK signaling pathway’ and ‘apoptosis’ (Fig. 2D).

Construction of PPI network and identification of hub genes

To further explore the promising molecular complexes responsible for the aberrant activation of alveolar macrophages in ARDS, the PPI network was constructed basing on the top 300 DEGs using the STRING database and visualized by Cytoscape software (Fig. 3A). Moreover, MCODE tool was applied to identify functional modules and the top 5 ranked clusters were showed in Fig. 3B.

Figure 3 Protein–protein interaction (PPI) network based on the top 300 DEGs.

(A) PPI network was constructed basing on the top 300 DEGs. The red icons represent up-regulated genes and the blue ones represent down-regulated genes. Furthermore, the connecting lines between the icons indicate potential interaction between different DEGs. (B) Module analysis of the DEGs enrolled in PPI network using MCODE method. The top five ranking clusters are shown according to MCODE score. (C) Functional enrichment analysis of genes in cluster 1. Likewise, the y axis and x axis represent enrichment terms and scores, respectively, and the color means p- value. (D) The top 10 hub genes are identified from PPI network using MCC method. The color represents MCC score. A darker red icon means higher MCC score and a lighter yellow one represents lower MCC score. (E) mRNA expression levels of CCL13, CCR2, FPR3 and PLAU in alveolar macrophages isolated from better prognosis subjects (VFDs > 12) and from poor prognosis individuals (VFDs <12).

Afterwards, we performed functional enrichment analysis for these promising functional clusters using Metascape. For cluster 1, the most enriched terms were ‘adenylate cyclase-modulating G protein-coupled receptor signaling pathway’ , ‘G alpha (i) signaling events’, ‘cellular calcium ion homeostasis’, ‘response to tumor necrosis factor’ and ‘Neuroactive ligand–receptor interaction’ (Fig. 3C). A potential involvement of cluster 2 was revealed in oncogenic MAPK signaling, blood coagulation, apoptosis and cellular response to organonitrogen compound. Meanwhile, it was predicted that genes within cluster 3 implicated in ‘protein processing’, ‘cellular component disassembly’ and ‘myeloid leukocyte activation’. Besides, enrichment analysis indicated a possible involvement in endocytosis process for cluster 4 and in neutrophil activation and cell adhesion process for cluster 5, respectively.

Basing on the constructed PPI network, the hub genes which may play a key role in ARDS pathogenesis were identified by MCC method and the top 10 hub genes were chosen for subsequent research and showed in Fig. 3D. Noticeably, the 10 hub genes were coincident with the genes within cluster 1 to a large extent indicating a crucial role of cluster 1 in ARDS onset and progress through Gi-protein-coupled receptor signaling and phosphatidylinositol-calcium second messenger activation.

Prognostic value of the hub genes

To explore the predictive capability of the 10 hub genes, we used t test to analyze the differences of transcript level between better (VFDs > 12) and poor prognosis group (VFDs < 12). Analysis of mRNA expression profiles from GSE116560 indicated that none of the 10 hub genes showed statistically significant difference of mRNA expression between better and poor prognosis group. However, CCL13 (p = 0.0558), CCR2 (p = 0.1578) and FPR3 (p = 0.1611) showed a mild trend of higher transcript level in poor prognosis subjects (Fig.3E). Meanwhile, a slight but not statistically significant over-expression of PLAU was observed in better outcome individuals (p = 0.2631) (Fig.3E). These findings implied that CCL13, CCR2, FPR3 and PLAU may play a crucial role in the onset and progression of ARDS, though no statistically significant result was concluded.

Establishment of cell model of LPS-induced ARDS

For the purpose of differentiation into a macrophage phenotype, THP-1 cells were incubated with PMA (50 ng/ml) for 2 days and observed by phase-contrast microscopy. Compared to untreated cells, PMA treatment remarkably enhanced the adherence of THP-1 cells and reduced the nucleocytoplasmic ratio and these changes were viewed as main features for macrophage differentiation (Fig. S1A). Flow cytometry analysis revealed that, compared to THP-1 cells with higher expression of surface marker CD14, THP-1-derived macrophages differentiated by PMA expressed higher levels of both CD14 and CD68 on surface (Fig. S1B). Mature THP-1-derived macrophages secrete TNFα in response to pro-inflammatory stimuli such as LPS (Park et al., 2007). To assess the pro-inflammatory response of THP-1-derived macrophages to LPS, cells were treated with LPS (1 µg/ml) for 12 h and the secretion of TNFα was evaluated by ELISA. Data indicated that fully-differentiated macrophages pre-treated with PMA secreted higher levels of TNFα in response to LPS (P < 0.05 versus LPS-stimulated THP-1 cells), whereas non-differentiated THP-1 cells showed negligible TNFα secretion in the presence of LPS stimulation (Fig. S1C). These data verified that THP-1 cells differentiated with PMA (50 ng/ml) can be used as a model of macrophages. After optimized THP-1 differentiation into macrophages, cells were challenged with LPS (1 µg/ml) for a varying time of 0, 1, 3, 6, 12 and 18 h and macrophage model of LPS-induced ARDS was successfully established.

Up-regulated expression of FPR3 and CCR2 in macrophage model of LPS-induced ARDS

Next, we sought to validate the expression trend of the hub genes obtained from mRNA microarray assay. Using GeneCards database (http://www.genecards.org/), we selected 6 hub genes (CCL13, FPR3, PLAU, CX3CR1, CXCL16 and CCR2) for further study mainly due to their potential involvement in chemotaxis-related signaling pathways. Afterwards, qRT-PCR was performed and the expression levels of CCL13, FPR3, PLAU, CX3CR1, CXCL16 and CCR2 in macrophage model of LPS-induced ARDS were detected. As shown in Fig. 4A, LPS stimulation induced significantly enhanced FPR3 expression in a time-dependent manner, which was coincident with the previous microarray data. Meanwhile, we also observed that CCR2 mRNA expression was dramatically elevated after LPS stimulation, though this trend was contradictory with that obtained from microarray assay. Afterwards, Western blot analysis was performed to examine the protein levels of FPR3 and CCR2. A substantial up-regulation of CCR2 protein on macrophages after LPS addition was detected. Likewise, LPS stimuli dramatically enhanced FPR3 protein level in a time-dependent manner (Fig. 4B). Moreover, flow cytometry indicated a significantly elevated expression of CCR2 on surface of PMA-differentiated macrophages after LPS stimulation (Fig. 4C). However, FPR3 was hardly detected on surface of THP-1-derived macrophages no matter LPS was administered or not (Fig. S2).

Figure 4 Expression levels of CCR2 and FPR3 in macrophage model of LPS-induced ARDS and primary alveolar macrophages from ARDS patients.

(A) Relative mRNA expressions of CCL13, CCR2, CX3CR1, CXCL16, FPR3 and PLAU on THP-1-derived macrophages in response to LPS were assessed using qRT-PCR. (B) The protein levels of CCR2 and FPR3 under a varying time of LPS treatment were evaluated by Western blot analysis. (C) CCR2 expression on surface of THP-1-derived macrophages after LPS treatment was detected by flow cytometry analysis. (D) Relative mRNA expressions of CCR2 and FPR3 on primary AMs collected from patients and volunteers were evaluated using qRT-PCR. (E) The protein levels of CCR2 and FPR3 in primary AMs were examined using Western blot assay. (F) CCR2 expression on surface of primary AMs was detected by flow cytometry analysis. Data are presented as mean ± SD and are representative of three independent experiments. ∗P < 0.05, by unpaired, 2-tailed Student’s t test.

Elevated expression of FPR3 and CCR2 in primary alveolar macrophages of ARDS patients

To further confirm the expression levels of FPR3 and CCR2 in primary alveolar macrophages of ARDS patients, BALF samples were collected from 6 ARDS patients and 4 healthy volunteers and primary alveolar macrophages were isolated using magnetic-activated cell sorting. qRT-PCR indicated that the expression of CCR2 was higher in primary AMs of ARDS patients than in those of healthy volunteers (P < 0.05 versus healthy volunteers). Likewise, an elevated FPR3 expression was detected in primary AMs from ARDS patients compared with healthy individuals (P < 0.05 versus healthy volunteers) (Fig. 4D). Moreover, we also observed a significant increase of FPR3 and CCR2 protein levels in primary AMs of ARDS patients (Fig. 4E). Besides, flow cytometry analysis revealed that expression of surface marker CCR2 was significantly elevated in primary AMs of ARDS patients (Fig. 4F) and FPR3 was barely detected (Fig. S3), which was in line with our previous findings in macrophage model of LPS-induced ARDS.

Collectively, these data confirmed the ectopic-expression of FPR3 and CCR2 in both macrophage model of LPS-induced ARDS and primary alveolar macrophages collected from ARDS patients. These observations suggested a possible participation of FPR3 and CCR2 in the initial inflammation response of alveolar macrophages in ARDS.

FPR3 and CCR2 down-regulation attenuated LPS-induced macrophage chemotaxis toward CCL2

Using in vitro transwell assay described above (Fig. 5A), we examined the ability of macrophage chemotaxis toward CCL2. LPS stimulation induced significantly enhanced macrophage migration toward CCL2 than toward culture medium (P < 0.05 versus PBS-treated macrophages) (Fig. 5B), which may correlate with LPS-induced over-expression of CCR2 on macrophages (Figs. 4A–4C). This was confirmed by an inhibitory effect of siRNA-induced CCR2 silence on LPS-triggered macrophage chemotaxis toward CCL2 (P <  0.05 versus LPS-stimulated macrophages) (Fig. 5B). Therefore, our findings suggested that CCL2/CCR2 axis may take part in the monocyte/macrophage activation and migration into alveolar spaces in early ARDS. Further, we assessed the biological effect of FPR3 on LPS-induced macrophage migration toward chemokine CCL2. As shown in Fig. 5B, LPS-induced increase in macrophage chemotaxis was strongly attenuated by siRNA-FPR3 transfection (P <  0.05 versus LPS-stimulated macrophages). However, neither FPR3 nor CCR2 silence affected the LPS-induced macrophage migration toward chemoattractant fMLP which is also known as a common ligand for FPR2 and FPR1 inducing chemotaxis (Fig. 5C).

Figure 5 Assessment of CCR2 and FPR3 in LPS-induced chemotaxis of macrophages in in vitro experiment.

(A)Diagrammatic illustration for the chemotaxis assay procedure: Upper:1 ×10 4 of wild-type or siRNA-transfected macrophages were cultured in serum-free medium with or without LPS. Lower: Culture medium with or without CCL2/fMLP. (B) LPS-induced macrophage chemotaxis toward chemokine CCL2 was assessed using a transwell assay as described in the Methods. (C) Likewise, LPS-triggered macrophage migration toward fMLP was evaluated as above. (D) The correlation between CCR2 and FPR3 expression was analyzed using pearson correlation analysis with expression data of GSE89953 and GSE116560. (E) Elevated protein expression of CCR2 induced by LPS stimulation was significantly attenuated by siRNA-FPR3 transfection. Data are presented as mean ± SD and are representative of 3 independent experiments. ∗P < 0.05, by unpaired, two-tailed Student’s t test.

Considering the possible link between FPR3 and CCR2 shown in Fig. 3B, pearson correlation analysis was carried out with mRNA expression profiles of GSE89953 and GSE116560. Interestingly, we noticed a strong association of CCR2 with FPR3 mRNA expression in alveolar macrophages of ARDS patients (Fig. 5D). Moreover, we conducted FPR3 silence by transient siRNA transfection and observed that LPS-induced over-expression of CCR2 on macrophages was significantly inhibited (Fig. 5E), while FPR3 protein level was not obviously affected by siRNA-CCR2 transfection (Fig. S4).

Discussion

In the past decades, several supportive therapy strategies for ARDS have been proposed to reduce mortality. Lung-protective ventilation, compared with conventional ventilation approach involving a higher tidal volume, results in a significant reduction of mortality mainly by preventing ventilator-associated lung injury and pro-inflammatory mediators release (Acute Respiratory Distress Syndrome Network et al., 2000). Likewise, other approaches such as a higher positive end-expiratory pressure (PEEP) (Fan et al., 2017), ventilation in prone position (Rhodes et al., 2017; Guérin et al., 2013), a conservative fluid-management (National Heart, Lung, and Blood Institute Acute Respiratory Distress Syndrome (ARDS) Clinical Trials Network et al., 2006) and neuromuscular blockage usage (Papazian et al., 2010) are associated with reduced ARDS mortality and currently recommended. In spite of these therapies described above, the mortality rate of ARDS has remained high. Besides, pharmacological approaches such as inhaled nitric oxide (Griffiths & Evans, 2005) and glucocorticoids (Steinberg et al., 2006) have failed to show survival benefit, which highlights the urgent need in the development of effective therapeutic strategies for ARDS.

In this study, 710 DEGs were screened out and the top 300 DEGs were chosen for further analysis. GO analysis showed that the biological processes of the DEGs were mainly associated with myeloid leukocyte activation, cell chemotaxis, regulation of inflammatory response, positive regulation of hydrolase activity, lymphocyte activation, positive regulation of defense response, leukocyte differentiation, response to wounding, organic hydroxy compound metabolic process, phagocytosis and regulation of cytokine production. In addition, we conducted KEGG analysis and revealed a main enrichment for these DEGs in Phagosome, Apoptosis, B cell receptor signaling pathway, Chemokine signaling pathway and PPAR signaling pathway. To prevent from failing to detect biological processes that are distributed across the entire gene network and subtle at the level of individual genes, GSEA was introduced to analyze the genome-wide mRNA expression data at the level of gene sets. Consistent with the previous GO and KEGG analysis using Metascape, GSEA highlighted biological programs of adenylate cyclase inhibiting G protein coupled receptor signaling pathway, macrophage migration, ERK1 and ERK2 cascade, cytokine-cytokine receptor interaction, PPAR signaling pathway, complement and coagulation cascades and JAK-STAT signaling pathway as the key transcriptional changes in activated AMs of ARDS.

WGCNA is an analytical method to identify gene co-expression networks basing on topological overlap and systematically analyze the relationship between gene modules and traits. We successfully identified eight co-expression gene modules and functional enrichment analysis was conducted for clustering genes of each module. Leukocyte activation, neutrophil chemotaxis, complement and coagulation cascades, cytokine-cytokine receptor interaction and chemokine signaling pathway were the most enriched terms, which coincided with findings acquired in previous analysis. These observations enhanced the concept that macrophage/monocyte chemotaxis toward alveolar spaces in the early-stage of ARDS may contribute to tissue damage and the underlying molecular mechanism may be adenylate cyclase inhibiting G protein coupled receptor signaling and phosphatidylinositol-calcium second messenger activation.

We managed to construct the PPI network basing on the DEGs and the key gene clusters were identified according to MCODE method. Using MCC method, we further screened out the hub genes which may take effect in the pathogenesis of ARDS. We noticed that the top 10 hub genes and the genes in cluster 1 overlapped to a large extent. This observation indicated a crucial role of cluster 1 in the excitation and accumulation of AMs in ARDS. Functional enrichment analysis of genes in cluster 1 re-emphasized Gi-protein-coupled receptor signaling events and phosphatidylinositol-calcium second messenger activation as the possible mechanism underlying the aberrant activation of AMs. We next utilized the mRNA expression matrix of GSE116560 to evaluate the predictive capability of the 10 hub genes. Regretfully, none of the 10 hub genes showed a significant difference of expression between better and poor prognosis subjects and this phenomenon may be due to a small sample size.

By comprehensive analysis with GeneCards online tool, we chose 6 hub genes for further research to explore their possible involvement in inflammation-associated signaling pathways and diseases. For this purpose, macrophage model of LPS-induced ARDS was correctly established. Using qRT-PCR and Western blot assay, we detected a significantly elevated mRNA and protein expression of FPR3 and CCR2 in macrophage model of LPS-induced ARDS. Furthermore, flow cytometry analysis revealed an elevated CCR2 expression on surface of THP-1-derived macrophages under LPS challenge. To further validate the expression trend of FPR3 and CCR2 obtained in macrophage model, we successfully isolated primary alveolar macrophages from ARDS patients and observed a similar elevated expression of FPR3 and CCR2 as expected. These findings convinced us of a potential role of CCR2 and FPR3 in macrophage activation and migration in ARDS.

As a main ligand for CCR2, CCL2 has been detected in BALF of ARDS patients and verified as a pivotal chemokine involved in regulating neutrophil migration (Williams et al., 2017).This suggests a possible involvement of CCL2/CCR2 axis in macrophage activation and accumulation, which was supported by an inhibitory effect of CCR2 down-regulation on LPS-induced macrophage chemotaxis toward CCL2. Likewise, we observed that siRNA-FPR3 transfection resulted in a substantially attenuated chemotaxis toward CCL2. However, neither FPR3 nor CCR2 silence had notable effect on LPS-triggered macrophage chemotaxis toward fMLP. These findings raised a hypothesis that FPR3 possibly interacted with CCL2/CCR2 axis by a certain undefined mechanism and affected CCL2-mediated macrophage chemotaxis. This hypothesis was supported by pearson correlation which indicated a positive and strong correlation between CCR2 and FPR3 expression in alveolar macrophages of ARDS. Moreover, CCR2 overexpression in macrophage model of LPS-induced ARDS was substantially inhibited by FPR3 silence. These observations suggested a possible interaction between FPR3 and CCL2/CCR2 axis in ARDS pathogenesis.

The impressive work conducted by Morrell and colleagues (Morrell et al., 2019; Morrell et al., 2018) uncovered a distinct pattern of transcriptional programs in alveolar macrophages. Based on research by Morell et al., we further comprehensively analyzed the mRNA expression data using a variety of analytic methods and successfully identified the candidate genes which potentially participated in ARDS pathogenesis. More than that, we confirmed a positive effect of FPR3 and CCR2 on CCL2-mediated recruitment of macrophages and the underlying molecular mechanism may be an unknown direct or indirect interaction of FPR3 with CCL2/CCR2 axis.

Conclusions

In conclusion, our findings strongly suggest that FPR3 and CCR2 are promising candidate genes which promote macrophage chemotaxis in early ARDS through a possible interaction between FPR3 and CCL2/CCR2 axis. Although further research is warranted, the evidence obtained so far improves our understanding on the underlying molecular mechanism and indicates that FPR3 and CCR2 might be novel therapeutic targets for ARDS treatment.

Supplemental Information

Supplemental Information 1 Establishment of macrophage model of LPS-induced ARDS

(A) Phenotypes of PMA-differentiated macrophages and non-differentiated THP-1 cells were observed by phase-contrast microscopy. (B) Surface marker CD14(monocyte) and CD68(macrophage) were detected by flow cytometry analysis. (C) The TNF α release in response to LPS was evaluated by ELISA assay.

Click here for additional data file.

Supplemental Information 2 FPR3 expression in macrophage model of LPS-induced ARDS was assessed by flow cytometry analysis

Click here for additional data file.

Supplemental Information 3 FPR3 expression on surface of primary alveolar macrophages was detected by flow cytometry analysis

Click here for additional data file.

Supplemental Information 4 LPS-induced elevation of FPR3 protein on macrophages was not obviously affected by siRNA-CCR2 transfection

Click here for additional data file.

Supplemental Information 5 The primer sequences of hub genes

The primer sequences of 6 hub genes including FPR3, CXCL16, CCL13, CX3CR1, PLAU and CCR2 are shown.

Click here for additional data file.

Supplemental Information 6 Raw pictures of Western blot PVDF membranes

Click here for additional data file.

Supplemental Information 7 Raw data for qRT-PCR&macrophage chemotaxis

Click here for additional data file.

Supplemental Information 8 Uncropped blots

Click here for additional data file.

Supplemental Information 9 Flow cytometry analysis

Click here for additional data file.

Supplemental Information 10 GSEA data: enriched GO terms in AMs

Click here for additional data file.

Supplemental Information 11 GSEA data: enriched KEGG terms in AMs

Click here for additional data file.

Supplemental Information 12 GSEA data: enriched GO terms in MONOs

Click here for additional data file.

Supplemental Information 13 GSEA data: enriched KEGG terms in MONOs

Click here for additional data file.

Additional Information and Declarations

Competing Interests

Author Contributions

Data Availability

The authors declare there are no competing interests.

Yong Mao, Wei Xu, Youguo Ying and Zonghe Qin performed the experiments, prepared figures and/or tables, and approved the final draft.

Xin Lv conceived and designed the experiments, analyzed the data, prepared figures and/or tables, authored or reviewed drafts of the paper, and approved the final draft.

Handi Liao performed the experiments, analyzed the data, prepared figures and/or tables, and approved the final draft.

Li Chen and Ya Liu analyzed the data, authored or reviewed drafts of the paper, and approved the final draft.

The following information was supplied regarding data availability:

The PVDF membrane pictures of Western blots, raw qT-PCR data, raw GSEA data and raw FCM data are available in the Supplementary Files.

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
