# Peer review of "Identification and validation of candidate genes dysregulated in alveolar macrophages of acute respiratory distress syndrome"

_PeerJ, doi:10.7717/peerj.12312_

## Round 0.1 · original submission · Major Revisions

The reviewers feel that although the topic is of interest, substantial revisions are needed. If you chose to resubmit, please be sure to include a letter describing how you have responded to each of the concerns.

Reviewer 1 ·

Basic reporting

In this manuscript, the authors identified several HUB genes thought to be related to the pathogenesis of ARDS. The chemokine receptor CCR2 and a formylpeptide receptor FPR3 were chosen because of their relatively higher level expression by macrophages as defined by data base search. The authors subsequently performed some in vitro tests using LPS-stimulated THP-1 monocytic cell line as a model of “ARDS macrophages”. The found increased expression of CCR2 and FPR3 by LPS stimulated THP-1 cells and siRNA silencing of FPR3 unexpectedly suppressed cell responses to the CCR2 ligand CCL2. Overall, this study contains potentially interesting contents, but there are some serious concerns.
Major concerns
1. The data base mining is interesting and well performed. But the manuscript is in short of experiments for validation of established theories. A major problem is the authors did not use primary macrophages from patients and control subjects to measure gene expression levels, cell surface receptor density and signaling pathways that may confirm the existence of HUB networks to govern the progression of the disease. Use only THP-1 cell line as “ARDS macrophages” is inadequate.
2. There is no mention of neutrophils, which are also one major cell component in the pathogenesis of ARDS, recruited by chemokines and cause tissue damage.
3. The role of FPR3 is not well addressed. How FPR3 controls macrophage responses to chemokine CCL2 remains unknown, for instance, whether FPR3 in the context of ARDS may encounter any “endogenous” agonist(s) that may stimulate the receptor to transduce signals that cross-talk directly with CCR2 or indirectly through yet to be identified pathways.

Minor concerns
1. It is conceptually erroneous to name fMLP(F) as a “chemokine”. Its receptors belong to a non-chemokine GPCR subfamily.
2. There are a number of typographical and grammatical errors to be rigorous corrected in future forms of the manuscript.

Experimental design

The data base mining is interesting and well performed. But the manuscript is in short of experiments for validation of established theories. A major problem is the authors did not use primary macrophages from patients and control subjects to measure gene expression levels, cell surface receptor density and signaling pathways that may confirm the existence of HUB networks to govern the progression of the disease. Use only THP-1 cell line as “ARDS macrophages” is inadequate.

Validity of the findings

Doubtful, although potentially interesting.

Additional comments

See above

Reviewer 2 ·

Basic reporting

Attention to language would benefit the manuscript. There is an overuse of the word “nowadays”; this term is not specific and not used properly.
Sentence structure and word usage is sometime not correct.
One example: “Despite the occurrence of several therapeutic strategies, the nowadays therapy for ARDS is disappointing” is a sentence that could be improved by giving specific details on current therapeutic strategies and rather than use the word “disappointing” use something like “ineffective” or “inadequate”.
Line 115: “By now” should be deleted/replaced… “Currently”

While there are some issues with certain parts of the manuscript as noted above, the Introduction contains good content, with an appropriate level of detail and background information.

The last section of the Results (lines 400-406) are written in a very confusing way and lack a summary statement.

Experimental design

This manuscript described a study using previously collected data from ARDS patients. These data sets examined lung alveolar macrophages and peripheral blood, with one data set being collected early during disease (day 1) and used as a prognostic metric by examining outcomes in the ICU. Overall, the details were described and referenced in sufficient detail.

The research question is focused on defining predictors of poor outcomes in ARDS patients and uses appropriate datasets.

To validate their findings using human gene expression datasets, they perform chemotaxis assays using THP-1 differentiated macrophages.

Validity of the findings

The authors merge to prior datasets to attempt to identify novel signaling networks that may be targeted to treat ARDS. Overall, data are accurately presented. The authors clearly qualify the findings where statistical significance is lacking, and appropriately temper conclusions.

The studies nicely build on human datasets and examine the biological consequence of CCR2 and FPR2 on chemotaxis.

The authors should provide more clear information on how their study goes beyond what was published using the previously published dataset and what was concluded and revealed in that manuscript.

Additional comments

The poor outcomes in ARDS patients is important and warrants investigation. The authors leverage datasets to examine predictors of poor outcomes and then attempt to validate their findings using in vitro assays and a monocyte cell line. Overall, the study is well-described and presented in the figures. There are several comments that should be addressed:

1. There are a number of gene networks that were described as being increased that are somewhat confusing, such as B cell receptor signaling, and raise questions about the purity of the monocyte/macrophage cells collected. This should be commented on.
2. G-protein coupled receptors emerged as a key potential player, though no significant differences were obtained in terms of outcomes (Figure 3E). It is not clear why/how these genes were chosen for follow-up. The networks depicted in panels A and B are not terribly easy to interpret and one suggestion is to show the expression data for all CCL13 receptors and CCR2 ligands.
3. Authors include CXCL16 data in Figure 4. This should be evaluated in the human datasets presented in Figure 3.
4. One of the key findings from the THP1 experiments is that silencing FPR3 seems to reduce CCR2 expression, though it is poorly summarized in the text. In general the authors do not provide much information or context to their interpretations of FPR3 and its role in ARDS.
5. The authors should clearly articulate how this analysis/re-analysis moves beyond what was published in PMID: 30990758

---

## Round 0.2 · Minor Revisions

While the manuscript is much improved, English editing is still needed and a few other minor points need to be addressed.

Reviewer 1 ·

Basic reporting

The authors made substantial effort to address the concerns expressed by this reviewer. The current version is now considered acceptable.

Experimental design

The approaches used by the authors to address the questions are reasonable.

Validity of the findings

The hypothesis and conclusions are reasonably made.

Additional comments

The authors have reasonably revised the manuscript.

Reviewer 2 ·

Basic reporting

Though the writing is much clearer in the revised submission, there are still issues with the manuscript in terms of language usage. There are spelling errors and incorrect grammar and punctuation.

Experimental design

The study builds upon a deposited gene expression data set. The deeper analysis was aimed at understanding the responses to acute respiratory distress syndrome and revealed the potential role for chemotactic signals. Overall, the research goal and the data analysis was described clearly.

Validity of the findings

The findings are clearly stated and presented. One concern is that the conclusions that CCR2 is important for pathogenesis is based on a correlation of it's expression and macrophage migration, and there is no evidence provided to conclude that CCR2 drives disease.

Additional comments

This resubmission addresses some of the original questions and criticisms, though there are still some concerns that should be addressed:
1. Writing and language usage needs editing.
2. The entire study provides correlative data only. The observation that CCR2 would mediate chemotaxis towards a CCL2 gradient is expected. There is no evidence that Ccr2 expression is actually "over-expressed" as stated on line 133, it's simply the expression after LPS challenge.
3. Data in Figure 4 are not convincing regarding the increase in CCR2 expression in ARDS patients. The MFI for CCR2 should be shown as the major difference really appears to be an increase in CD68+ cells. This should be discussed and is not clearly stated or summarized.

---

## Round 0.3 · accepted · Accept

Thank you for submitting your work to PeerJ. We are pleased to now accept the manuscript and we hope that you will consider PeerJ for future manuscripts from your lab.